# Synthesizing Policies That Account For Human Execution Errors Caused By State Aliasing In Markov Decision Processes

## Abstract

When humans are given a policy to execute, we expect there to be erroneous executions and delays due to possible confusions in identifying a state. So if an algorithm were to compute a policy for a human to execute, it ought to consider these in its decision. An optimal policy that is poorly executed can be much worse than a suboptimal policy that is executed faithfully and faster. In this paper, we consider these problems of delays and erroneous execution when computing policies for humans that would act in a domain modeled by a Markov Decision Process (MDP). We present an algorithm to search for such policies and show experimental results in a Warehouse Worker domain and Gridworld domain. We also present human studies to show how our assumptions translate to real-world behavior.

## Introduction

Markov Decision Processes (MDPs) have been used extensively in many applications([Boucherie and Van Dijk 2017],[Hu and Yue 2007],[White 1993]) but what if the agent that has to act in such a scenario is a human, the optimal policy maybe too complex to reasonably expect a human to execute it accurately or quickly. Our cognitive and perceptual limitations may result in mistakes, such as confusing similar states. We may also take longer to execute an optimal policy since it requires more cognitive effort to discern between similar states, which is necessary when the policy for those states are different. A sub-optimal but simpler policy that can be more faithfully and quickly executed can be preferable in some scenarios. There is precedent for preferring simpler policies in the medical literature; one example of this is the Apgar score[AmericanAcademyOfPediatrics 2006]. It is a policy that relies on a simple scoring method to determine what action to take with newborn babies. Doctors and nurses are taught a simple scoring system on few easily measured signals to determine the health(state) of the baby and act according to this single score. A more complex policy that is conditioned on more or granular measurements and prior states could result in costly mistakes.

In our paper, we specifically consider the problem of confusing similar states, i.e., state-aliasing, and how that could affect the value of a policy in Markov Decision Processes

(MDP). We work with the assumption that a policy which uses the same action across similar states is easier to follow or execute (which we show in human studies), and that similar states can potentially be confused with each other (state-aliasing), especially under time pressure or other stressors. This state aliasing can lead to errors in execution due to misidentifying states. It can also result in delays in execution when similar states have different actions in the policy; if the actions were the same, then there is no need to wait and discern the states properly.

Our work in this paper is connected to prior work by [Whitehead and Lin 1995], which considers state-aliasing through errors in perception (robot sensors). However, their objective is to improve the sensing policy (active perception) so as to have a better internal representation for the policy execution. This would add to the difficulty of following the policy and we cannot expect the user to consistently compute accurate posterior likelihoods given a sensing process. Instead, we take the likelihood of the human agent (mis)classifying states as an input. These classification likelihoods can be empirically determined through evaluating the human; we will discuss this more shortly.

To illustrate the problem, let us consider a simple version of a *Warehouse Worker* domain. In this domain, a worker is at the end of a conveyor belt and customer orders of different sizes arrive. The human has to decide the size of the box needed (small, medium, or large). If (for example) the difference in cost of box sizes is very small, then the simplest policy is to always use the large box. This would save on delays to decide the right action. If the policy actions are easily decided, then more orders are completed (greater throughput), and the company gets more revenue. More importantly, the cost of erroneous execution– trying to put a medium-sized order in a small box– is avoided. If one were to ignore policy execution errors and delays, the optimal policy computed for the original MDP could actually be suboptimal when it is executed by the human due to the delays and errors in execution. Such execution errors can especially be pronounced in high-stress situations, which tend to cause people to miss perceptual cues (Tunnel vision/Tunneling hypothesis), and poor cognitive performance [Staal 2004].

In this paper, we formally define the problem of computing a policy for a State-Aliased MDP (SAMDP). In our definition, we describe how to model two effects of state-

aliasing on human policy execution; the likelihood of inaction (delay) and the likelihood of erroneous execution. We also quantify the notion of policy-confusion likelihood. We then present a modified policy-iteration algorithm that searches for policies that optimize for value while considering delays and erroneous execution. Our algorithm also supports weighting the search to look for those with lower policy confusion likelihood; such policies can be easier for humans to follow. We show experimental results for our approach on two domains; Gridworld and Warehouse Worker domain. We also present the results of our human studies which compares the execution of simple and difficult policies (higher likelihood of policy confusion) and show how our assumptions translate to real-world behavior. Lastly, we make our codebase available for future research, and it supports defining any discrete state, discrete action MDP through CSV files (Comma Separated Values); these can be conveniently edited in any freely available spreadsheet software.

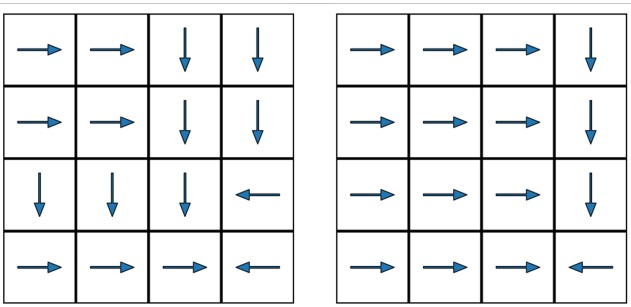

Figure 1: High value policies corresponding to different policy confusion scores; the policy on the left has a higher confusion score.

## Problem Definition

The problem of generating policies for a State-Aliased MDP (SAMDP) is defined by the tuple $< S, A, T, r, \gamma, \phi, p >$. Each of the terms are defined as follows:

- $S$ is the set of states in the domain;

- $A$ is the set of actions including $a_\varnothing$ which is a state reidentification action

- $T : S \times A \times S \to [0, 1]$ is the transition function that outputs the likelihood of transition from one state to a successor state after an action. This includes the reidentification action transitions.

- $r : S \times A \to \mathbf{R}$ is the reward function. This includes the reward associated to reidentification actions.

- $\gamma$ is the discount factor

- $\phi : S \times S \to [0, 1]$ is the likelihood of classifying the first state as the second .

- $p : S \to [0, 1]$ is the likelihood of starting in a particular state.

The objective is to output is a policy $\pi : S \to A$ that seeks to maximize the policy value (equation 2) while mitigating the delays by keeping the policy confusion score (equation - 3). One must account for the classification likelihoods when computing the policy since the policy that is actually executed (and the value) is affected by them. The Classification likelihood ($\phi$) is the pivotal factor in this problem. If there was no classification of states (state aliasing), then it is a standard MDP, and policy iteration would solve it.

The classification likelihood can lead to policy execution errors. This can happen when the (mis)classification of states causes the actions of the aliased states to be executed in the current state. Formally, the probability of an action in a state for a given policy after accounting for the classification likelihoods is defined by Equation 1.

$$\pi_\phi(s, a, \pi) = \sum_{s' \in S} \phi(s, s') * \pi(s', a) \qquad (1)$$

where $\pi(s, a)$ returns the likelihood of the action in the input state for the policy $\pi$, and $\pi_\phi$ is the policy after consider $\phi$. Due to this effect on policy, the problem setting becomes non-markovian as the action and transitions in one state is affected by that of another state. The value after the policy is affected by state classification likelihoods is defined as:

$$V_\phi(\pi, p) = \sum_{s \in S} p(s) * V(\pi_\phi, s) \qquad (2)$$

where $V : \pi \times S \to \mathbf{R}$ is the value of the state for the policy $\pi_\phi$; $\pi_\phi$ is the resultant policy after the effects of state-aliasing are applied to the input policy $pi$, as defined in Equation 1.

The state classification likelihood also determines policy confusion score, where the confusion score is formalized as:

$$CS(\pi, p) = \frac{1}{|S|} * \sum_{s_1 \in S} \sum_{s_2 \in S} \phi(s_1, s_2) * 1[\pi(s_1) \neq \pi(s_2)] \qquad (3)$$

The confusion score will lie in the range $[0, 1]$ where a score of 0 implies all similar states have exactly the same action. A score of 1 can only happen if a state is always mistaken for another state and the policy mismatches.

The reasoning for this quantification of confusion is as follows; if any pair of states can be confused with each other, but the action in the policy is different, then it adds to the likelihood of policy confusion. How much it adds to confusion is determined by the likelihood of confusing the two states, given by $\phi$, and the policy given to the human $\pi$; two very similar states with different actions adds more to the confusion score than two less similar states with different actions (less likely to confuse the two actions). So the classification likelihood is used as the weight when considering each pair of states whose actions in the policy do not match. For an example of policies that have different policy confusion scores but comparable values, see the gridworld example in image 1 where the sole reward is obtained by transitioning into the bottom-right grid. Note that we do not assume that the classification-likelihood ($\phi$) is symmetric; in some settings the state identification maybe biased.

# Effects of State Aliasing On Policy Execution By Humans

As mentioned, when state-aliasing is a problem, there are two effects on policy execution that we can model; policy execution delay and incorrect execution. Incorrect execution occurs when the human chooses the wrong action because the state was confused with another, and the action of the confused state was executed. This is as described in Equation 1.

The other effect is the action delay. After observing the real state, the human might infer that it is the (most likely) state $s_1$; this can be thought of as the maximum aposteriori probability (MAP) state in the human's mind after seeing the real state in the environment. However, the human might not be certain and think that it could be another state $s_2$ as well. If the actions are different in the policy for these two states, then one might spend additional time observing the environment again rather than act, resulting in a delay. If the actions were the same in the possible states, then the human can act without further delay. For example, in the Warehouse Worker domain, the policy is that all customer orders can be put in the large boxes. So, regardless of the workers confusion about the order size, they can act without further state-resolution and avoid delays.

We model the likelihood of delay in policy execution as increasing with the number of states that the current state could be confused with, and have different actions in the policy. We model this delay as a special action that represents the agent reidentifying the state. This is due confusion arising from uncertainty about the state. We assume that the human knows the policy or has access to the policy via a chart or device, so there is no consideration for forgetting the policy. The likelihood of reidentification action (and thus delay in policy execution) is determined by the policy and state confusion likelihoods. We define this in Equation 4.

$$p(a_\emptyset, s) = \sum_{i \in \{1...|S|\}} \phi(s_i, s) * \sum_{j \in \{i+1...|S|\}} \phi(s_j, s) * \\ 1[\pi(s_i) \neq \pi(s_j)] \quad (4)$$

This is the likelihood of one reidentification action. The likelihood of two reidentification actions is the product of two probabilities, and so forth with the likelihood of many successive reidentification actions becoming geometrically smaller. The worst case for a state in a domain is when a state is confused with every other state with equal probability. If there are only 2 states, with equal likelihood of confusion (50%) and different actions in the policy for these states, then this confusion likelihood is 25%; this is because there is only one case for confusion (thinking it could be state 1 or state 2). If a state can be confused with 3 other states with equal likelihood of confusion, and all 3 have different actions in their policy, then the confusion likelihood is 33%. As we increase the number of states that are equally likely to be confused and have different actions in policy, the confusion likelihood approaches 50% asymptotically (absolute worst case).

This confusion as to how to act because of state uncertainty leads to taking a reclassification/reidentification step, and thus causes a delay. This delay effect is worth considering if the time taken to identify a state, is comparable to the time to act. When this is true, the rewards for actions ought to be scaled accordingly. This can be done as in Equation 5.

$$\hat{r}(s,a) = r(s,a) * \gamma^{\lfloor t(s,a)/T_{min}-1 \rfloor} \quad (5)$$

where $T_{min}$ is the length of the shortest action, or the average time taken to identify a state, and $t(s,a)$ is the time taken by an action in a state. For example, if the average time to identify a state is 2 seconds, the discount factor is 0.9, and an action is 6 seconds with a reward of 10, then the updated reward is $10 * 0.9^{\lceil 6/2 - 1 \rceil} = 8.1$ . What this is says is that the action goes through intermediary states with reward 0, and each step is of time $T_{min}$. If actions take variable amounts of time, then the expected discounted reward should be used. This is comparable to the average reward rate described in [Das et al. 1999] in that the reward is scaled proportional to the time time. In our version, the action is for a fixed number of steps and is discounted. We describe this scaling of reward for the sake of completeness. In this work, we assume the input rewards are appropriately scaled since the reidentification(delay) cost for each state is part of the input specification.

If, however, actions take much longer than the time to identify a state, then the delay due to state identification could be ignored and just focus on erroneous execution. For emergency medical procedures such as a policy for crash cart [Martin-Cua 2018] usage, all actions are quick, and delay due to state identification is costly and should be factored in. In our methodology (and associated codebase) we provide the means to turn off consideration of the delay effect in our algorithm if that is the right choice for the problem by setting $p(a_\emptyset, s) = 0$ for all states which is otherwise computed as in Equation 4. Note that the dynamics of the reidentification action are defined by the domain, and maybe as simple as just staying in the same state.

## Policy Computation Algorithm for SAMDP

Finding an optimal solution to the SAMDP problem is challenging since the problem is both non-markovian and requires optimizing two sometimes opposing objectives; namely the policy value and confusion score. To handle this, we adopt a modified policy iteration approach that we call Global Value Policy Iteration (GVPI). The confusion score of a policy is factored into the value computation by the reidentification actions which occur more often in complex policies. When selecting actions during policy iteration, we need to consider not just the local value effect, but also the effect on the value of other states. This is because the possible state misidentification couples the policy of different states; this means the policy in one state affects the policy in another, and so affects the value of other states (possibly negatively). This can lead to update loops and never converge. So at each step of policy iteration, we consider the average value of over all states as the measure by which we update the policy.

When we evaluate a policy change in GVPI, we first compute the likelihood of reidentification actions for each state after the policy change. Then we compute the state transition likelihoods for that policy (including the reidentification action transitions), and compute the corresponding Markov Reward Process (MRP)[Ibe 2013]. Using this MRP we compute the value for all state using a closed form computation (will discuss shortly). We compute a score over the values for all states and use that to choose the action in policy iteration. Since we consider all state values, we dubbed this "Global-Value Policy Iteration". We did try a policy gradient approach as well (search over the space of soft policies), and found GVPI to perform better for our experiments. We now explain our approach in detail.

## Computing Reidentification Action Likelihoods

In each policy iteration step of GVPI, we start with a deterministic policy. First, we account for the delay due to policy confusion. We compute this reidentification action likelihood as in Equation 4. The remaining probability $1 - p(a_\emptyset)$ is the likelihood of the human acting. Then we account for erroneous execution by considering the probability that the incorrect state was inferred. So the likelihood of an action being executed is defined by Equation 1. This gives us the updated policy $\pi_\phi$ that accounts for delays and erroneous execution.

## Translating To The Equivalent MRP

After computing the updated policy, we compute the equivalent Markov Reward Process (MRP) associated to that policy; this is done by computing the transition likelihoods between ordered pairs of states based on the policy. The reward for each state is the reward for each action taken from that state, multiplied by the likelihood of that action being taken. This includes the reidentification action and the associated reward (cost) of that action in that state; it may not always be that reidentification action results in staying in the same state.

The reason we transform it to an MRP is that it allows us to exactly compute the values of all states in one closed form computation (Equation 6).

$$\vec{v_s} = (I - \gamma * P_{ss'})^{-1} * \vec{r_s} \qquad (6)$$

Where $I$ is the identity matrix, and $P_{ss'}$ is the probability of transition between two states, which is defined by the policy.

## GVPI Search Process

After computing the value of the states, we do not just sum or take the average of the state values, rather we consider the average of the inverse of state values as in Equation 7.

$$ps(\pi, p) = \sum_{s \in S} p(s) * \frac{1}{V_{\pi_\phi}(s) + 1} \qquad (7)$$

where $\pi_\phi$ is the policy derived from the input policy $\pi$ after applying the effects of state aliasing as described in Equation 1. $V(\pi_\phi, s)$ is the value of this policy after applying the reidentification action likelihoods in Equation 4.

With this score, GVPI iteratively proceeds to minimize this score, it does not maximize like in policy iteration since we have taken the inverse of the value. We chose to score actions this way because we observed that optimizing for the sum of state values can cause the policy iteration search to ignore the value of some states in favor of high value states since the overall sum is greater. In some problems, this maybe acceptable. But for some others, such as in gridworld, it produces more policies where the goal state is not always reachable from all states, which is undesirable. This score helps improve the state values of all states more uniformly; increasing the value of a state with value 0 contributes more to reducing the score (which is to be minimized) than improving a state with a higher value.

As one might intuit, this policy iteration search is not guaranteed to find the optimal policy for the expected value. Rather it is a means to generate a set of good policies that optimize for expected value and account for delays in execution. Repeated random restarts help find better policies, and in each attempt it will stop when the policy can no longer be changed to improve the score. This is as opposed to using standard policy iteration which could end up in infinite loops for SAMDPs. Since SAMDPs are non-markovian and the state policies are coupled, the policy update in one state could reduce the value of another state by changing it's policy. The policy iteration procedure could get stuck updating back and forth between coupled states, which we saw when we tried to solve SAMDPs with policy iteration.

## Trading Value and Confusion

Thus far we have not discussed how to explicitly penalize the search process for policy confusion. By incorporating a reidentification action into the execution when policies are confusing (similar states, different actions), the policy search automatically goes towards simpler policies unless the reward is sufficiently higher to merit additional risk of confusion. If one is interested in searching for even simpler policies, our methodology also supports pushing the search process to look for policies of lower confusion score, which will often come at the expense of value. The policy score is updated to allow this as in Equation 8

$$ps2(\pi, p) = (1 - \omega) * ps1(\pi, p) + \omega * CS(\pi, p) \qquad (8)$$

where CS is the confusion score from Equation 3 and $\omega$ is a hyperparameter between $[0, 1]$ which determines the emphasis given to reducing confusion, where 1 means the search will focus purely on reducing confusion. Additional scaling of the two scores was not helpful because the ps1 score is already in the range $[0, 1]$, and so too is the confusion score.

For an example of the kind of simple policies found by GVPI for a gridworld setting, see Figure 8. The policy on the right is the simpler policy output by GVPI. The policy on the left is the optimal policy without considering state-aliasing, that is output by standard policy iteration for the original MDP. The left policy has a lower expected value in SAMDP, and a higher policy confusion score.

## Obtaining the Classification Likelihood Matrix

The likelihood of two states getting misidentified would be affected by the domain features being used, and a person's perceptual capabilities. In this paper we take the state classification probabilities as input.

In an actual application, the probability of state classification (also misclassification) may need to be empirically determined. For example, in the warehouse worker domain an employee could be tested to see how they classify packages by a supervisor. This can be used to compute the classification likelihoods. When determining these likelihoods, the human should be asked to classify within some desired time cutoff or as quickly as possible. The classification likelihoods would be affected by the time allowed to identify a state for that domain. Then based on classification likelihoods, a policy maybe tailored specifically to that person. This data may also be averaged over groups of people, and a standard policy could be developed.

Alternatively, the state classification likelihood could be defined by normalized state similarities. This is reasonable if a similarity function that reflects human perception is available for the domain. We take such an approach for our human studies which will be discussed.

## Experiments and Results

We tested our algorithm on two domains; Warehouse Worker and Gridworld. The discount factor was set to $\gamma = 0.9$. We varied the weight for reducing confusion ($\omega$) between $[0, 1]$ in increments of $0.1$. For each setting, we ran the GVPI search 10 times. The results can be consistently reproduced from our codebase since any variability (like randomness in initialization) in the program is controlled by a seed parameter. This is set to 0.

### Warehouse Worker Domain Setup

In the Warehouse Worker domain, a worker stands at the end of a conveyor belt on which customer orders are sent. The customer orders comprises of a group of products. Each order needs to be put into a small, medium, or large box. Additionally, the worker has to decide if bubble wrap is necessary for the products in the order.

The states in this domain is what kind of an order a set of products actually is, and there is an associated correct action for each order type. The state and action sets are defined by the cartesian product of the set of box sizes $\{small, medium, large\}$, and if bubble wrap is needed $\{wrap, no\_wrap\}$. For example a set of glass items could be a small order that requires a small box with bubblewrap, $small \times wrap$. When a worker sees an order, it is not always apparent what box size is needed. For some orders, they may mistake a small order for a medium sized one or vice versa. Additionally, due to the diversity of products, the worker has no idea which products actually need bubble wrap or not. For example there maybe tempered (hardened) glass products that do not need bubble wrap, but the worker might not know this.

In our conceptualization of this domain, after the worker goes through basic training in the warehouse, the worker

is evaluated by the supervisor to evaluate how they classify orders. This becomes the classification likelihoods for $\phi$. Based on this, a policy is developed for the worker by considering the company's average estimates for the money made per order when using different types of packaging (reward specification), and the likelihood of order types.

We now detail the domain configuration settings we used in our experiments for the warehouse domain. The classification likelihoods are shown in Table 1. The likelihood of a worker confusing any small order with any medium sized order or vice versa is about $16\%$, and is the same for misclassifying any medium with any large order. The likelihood of confusing a small order with a large order is less than $1\%$. The likelihood of the worker correctly determining if bubblewrap is needed is $50\%$ across all order sizes; there are so many products that their accuracy for determining if bubble wrap is needed is random.

| | l | l × w | m | m × w | s | s × w |
|---|---|---|---|---|---|---|
| l | 32.68% | 32.68% | 12.50% | 12.50% | 0.98% | 0.98% |
| l × w | 32.68% | 32.68% | 12.50% | 12.50% | 0.98% | 0.98% |
| m | 16.34% | 16.34% | 25.00% | 25.00% | 16.34% | 16.34% |
| m × w | 16.34% | 16.34% | 25.00% | 25.00% | 16.34% | 16.34% |
| s | 0.98% | 0.98% | 12.50% | 12.50% | 32.68% | 32.68% |
| s × w | 0.98% | 0.98% | 12.50% | 12.50% | 32.68% | 32.68% |

Table 1: Classification Likelihood Matrix ($\phi$) for Warehouse Worker Domain, where (s,m,l) stands for (small,medium,large) and "w" means bubblewrap needed.

If the worker tries to put a medium sized order in a small box, the action will fail, and they will stay in the same state. Using any box size smaller than necessary will fail, but any larger size box will work. A successful action will transition to the next order (state) based on the probability of different types of orders. For our experiments we used a uniform distribution of customer orders (states). Lastly, the reward for using the exact action for an order is 1. The reduction in reward if a larger box is used is $-0.1$, and a further reduction of $-0.1$ is incurred if bubblewrap was needed but wasn't used. If bubble is used but not needed, there is no reduction in reward; we consider that the cost of bubblewrap in comparison to the monetary reward for a completing an order is negligible.

### Gridworld Experimental Setup

For our experiments in Gridworld, we used a 10x10 grid (100 states). The actions include moving up, down, left and right. The transitions are deterministic. The likelihood of confusing a grid position with another grid state is determined by the L1 distance as defined in Equation 9. This results in neighboring grid states being much more likely to get confused with the current state than those further away. Taking an invalid action, such as moving up from the top row of the grid results in no motion. The agent would get a reward of 100 for transitioning into the goal state in the bottom-right corner.

$$\phi_{grid}(s, s') = \frac{L1(s, s')^2}{\sum_{s'' \in S} L1(s, s'')^2} \quad (9)$$

## Results

The Expected Value of policies after accounting for delays and erroneous execution are shown in Figures 4 and 2. The box plot shows the upper and lower quartiles, and the circles represent outliers. The trendline connects the median values from each setting of $\omega$. The first setting with a "*" represents the optimal policies discovered policy iteration in the original MDP. We do 30 random restarts to get a set of optimal policies for the original MDP. The expected value of these "MDP-optimal" policies are not optimal after accounting for delays and errors in execution in the SAMDP. The policies found by GVPI are often better, especially for lower settings of $\omega$ for both domains. In both domains as $\omega$ increases the expected value goes down as expected. The expected value for gridworld may seem low but that is expected since it is averaged over 100 states with only 1 state having all the reward. Add to this the reward discounting, delays and erroneous execution, the values computed are smaller than one might expect. We verified this by hardcoding the known optimal policy which accounts for delays and execution errors, and it's expected value averaged over all states is 0.34. Our experiments can be easily verified/reproduced with our code which we will provide. All experiments are controlled by a random number seed and so is consistently reproducible.

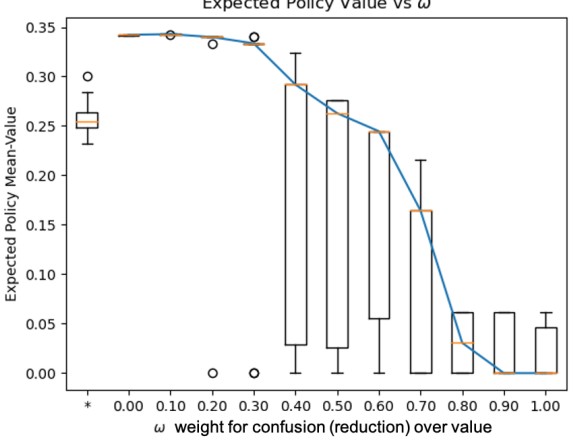

Figure 2: Expected Value of policies generated with varying $\omega$ in Gridworld

The confusion score of policies generated by varying $\omega$ are shown in Figures 3 and 5 for gridworld and warehouse domain respectively. To interpret these correctly, please keep in mind that the maximum confusion score a policy can have is 1, by our definition in Equation 3. In the confusion graphs of both domains, one will notice that the policy confusion is already quite low even with $\omega = 0$. This is because the effects of policy confusion is already folded into the computation of value in GVPI through the delay effect and erroneous execution. A simpler policy would have less of both,

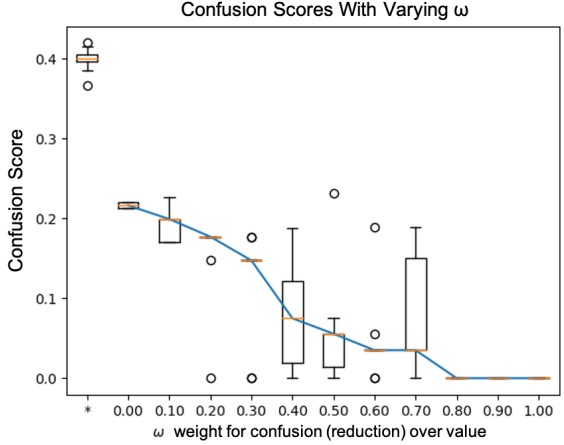

Figure 3: Confusion Score of policies generated with varying $\omega$ in Gridworld

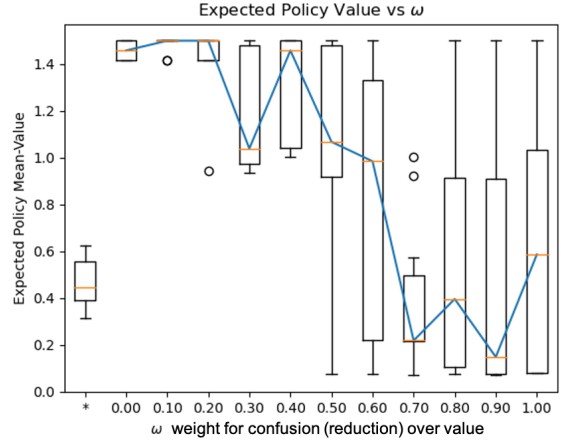

Figure 4: Expected Value of policies generated with varying $\omega$ in Warehouse domain

and so it is naturally preferred during the search. Increasing $\omega$ only serves to push the search even more towards simpler policies.

For Gridworld, the median confusion value of the policies decreases more gracefully with increasing $\omega$. We think this is likely due to the spread and availability of policies with different tradeoffs between value and confusion. This is not so in the Warehouse domain. Increasing the weight for confusion only results in it getting stuck at worse local optima in the search. We think this is due to the sparsity of policies. Also, note that the absolute value of the policy confusion is still low; the worst it does is 0.175 and the maximum confusion a policy can achieve in this domain is 1.0. In comparison, the optimal-in-MDP policies have a much higher(worse) confusion score, and their corresponding range of expected values in the SAMDP is low (Figure 4.

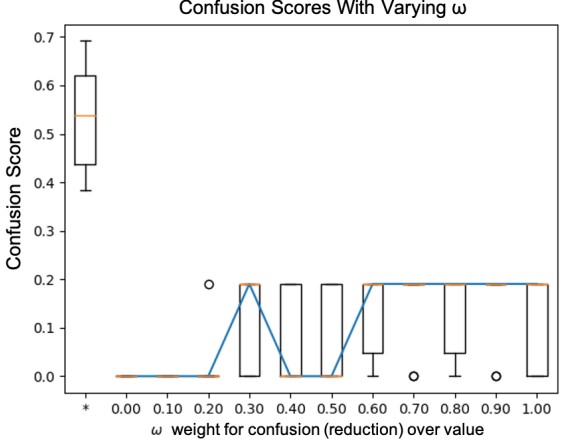

Figure 5: Confusion Score of policies generated with varying $\omega$ in Warehouse domain

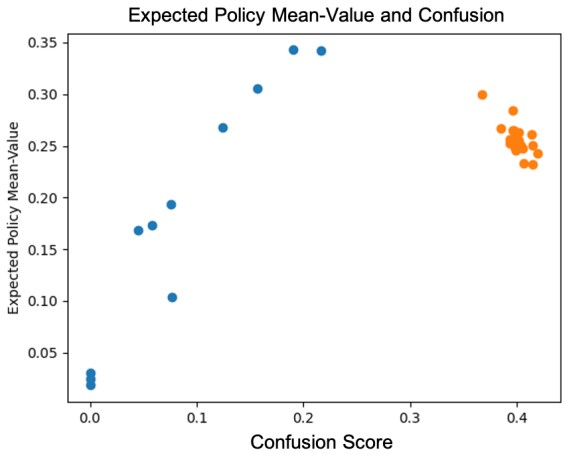

Figure 6: Expected Value and Confusion of policies generated during search in Gridworld

In both domains, we see high variance for expected value and confusion, especially in the middle range of $\omega$ values, which corresponds to the tradeoff difficulty during the search. Lastly, we show the expected value and confusion of policies in one plot for each of the domains in Figures 6 and 7 respectively. The blue dots represent policies discovered by GVPI, and the orange dots are the optimal-MDP policies. This is also to show that GVPI explores the space of tradeoffs between expected value and confusion.

## Human Studies

We conducted a human study to test the hypothesis that the execution performance of humans using a simple policy is higher in contrast to when a difficult (higher likelihood of confusion) policy is given. We gamified the study by asking each participant to execute a policy as given in Figure 8 by matching a displayed color to the appropriate arrow di-

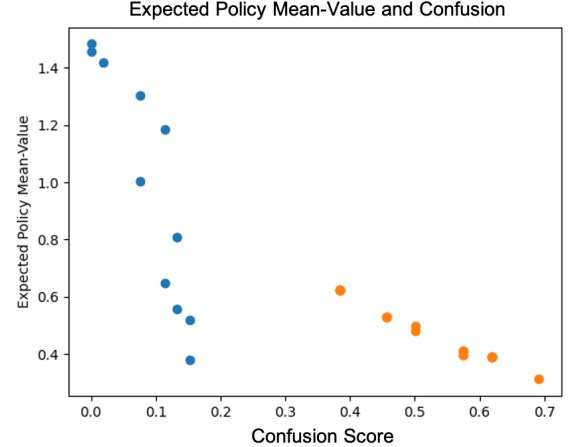

Figure 7: Expected Value and Confusion of policies generated during search in Warehouse domain

rection and maximize their score. The same of participants repeated the game twice; for a difficult policy and for a simple policy. The correct policy was always displayed on the side, so they did not have to memorize it, only follow it. Our objective was only to measure the number of actions, number of correct actions and error rate. These measures reflect how well the human can follow a given policy.

Note that some color states are intentionally visually similar to other color states to cause state-aliasing. The participants were filtered by their ability to distinguish between different colors so that they could execute both difficult and simple policies. The table 2 shows results for the 41 participants in this study.

We wanted to see if the number of actions executed was greater with the simpler policy; since less confusion likelihood should imply fewer delays. We are especially the number of correct actions (throughput). We also wanted to check if the rate of errors was lower. Since data from the two settings of simple and difficult policies may have unequal variances, we used a Welch's t-test (one-tail) to evaluate the results. We used the implementation in the Scipy python library [Virtanen et al. 2020]

For the total number of actions executed by a participant, we can reject the null hypothesis that the number of actions executed is the same or lower with the simpler policy than the difficult policy; the one-tailed T-test gave a p-value of $< .00001$. For the second hypothesis, that a simpler policy yields a higher number of correctly executed actions, a one-tailed T-test gave a p-value of $< .00001$. So with a very low likelihood of error, we can say our hypothesis held good in this human study. The results are clearly significant at p $< 0.05$.

We are also interested in the *rate* of errors when executing the simple versus difficult policies. We wanted to show that the rate of errors in a difficult policy is more than that of a simple policy. A one-tail T-test for the rate of errors (number of errors/total attempts) was significant only at a p-value of

0.065. So while we have good reason to believe this to be the case, we cannot say with confidence (p-value $< 0.05$) that the rate of errors is definitely lower. There might have been other factors affecting the rate of errors that we had not considered, such as the speed of execution of the simpler policy. One possible effect is that when the participants were acting very fast with the simpler policy, the likelihood of errors went up.

We also note that our GVPI algorithm consistently outputs the simpler policy shown in Figure 8 for a simple MDP corresponding to the human studies. In this MDP, the rewards are 1.0 for the correct action(s) in the simple policy, 1.1 for the actions that are different in the difficult policy (see Figure 8), and 0 for all other actions. Transitions to successor states are independent of the action and equally likely; we needed this to test policy execution uniformly across all states. The discount factor $\gamma$ was 0.9, and we modeled the state classification likelihoods such that similar color states were equally likely (50%) to be confused as each other. The simpler policy output by GVPI is desirable since the additional reward for choosing the optimal action is small compared to the loss that could be incurred due to delays and incorrect execution.

Overall, our human studies show that giving a simpler policy reduces the delays (higher number of actions executed) and increases the throughput (number of correct actions). This translates to more rewards accrued. The rate of incorrect actions was not conclusively shown to be lower, even if the data gives us good reason to think so. There could be more factors that affected the execution, such as the rate of policy execution. Running each trial for longer might give us more conclusive data.

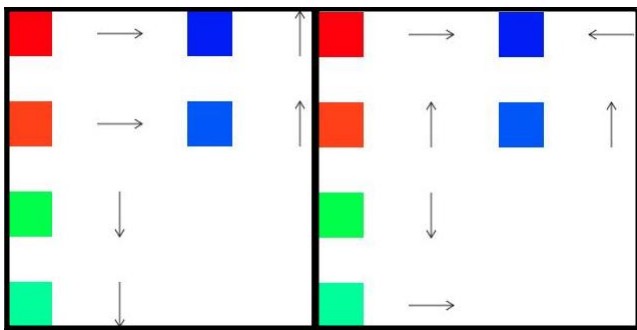

Figure 8: The simple policy (left), and difficult policy (right) given to users to execute

|  | Simple Policy $(\mu, \sigma)$ | Difficult Policy $(\mu, \sigma)$ |
| --- | --- | --- |
| Correct Attempts | 26.34, 5.62 | 17.95, 3.77 |
| Total Attempts | 28.12, 6.13 | 19.68, 3.82 |

Table 2: Mean and standard deviation for the number of correct actions and total actions executed for the simple and difficult policies

## Related Work

In the scientific literature on teaching humans a policy to execute, there is a strong precedent for giving simpler policies to humans. As mentioned, the Apgar score [AmericanAcademyOfPediatrics 2006] is one such example from the medical literature. With the Apgar score, the doctors can score how well a baby endured the birthing process and, based on the score, determine what subsequent steps are needed. It is a scoring process based on a few features and is recomputed at 5-minute intervals to check the baby's health periodically. Rather than have one very complicated procedure, a simpler, reliable computation is done more often. This idea was taken further in the work "Super Sparse Linear Integer Models (SLIM)" by [Ustun and Rudin 2016] in which the authors build sparse linear models with emphasis on smaller integer weights because they make computation by humans easier and more reliable.

We are similarly motivated to compute simpler policies for MDPs that account for our cognitive limitations, and errors we may make. Specifically, we consider how humans can confuse similar states, especially under duress or time constraints, and how it can be easier to work with a policy that maps similar states to similar actions. [Lage et al. 2019] works with a similar assumption, and proposes an Imitation Learning (IL) based summary extraction that uses a Gaussian Random Field model [Zhu, Lafferty, and Ghahramani 2003] for human policy extrapolation. They considered that people use the similarity between states for generalizing policy summaries to states that were not part of the summary. For one of their domains, they reported that 78% of their participants used state similarity based policy-summary reconstruction. The authors also argued that state similarity could have an effect regardless of the objective or reward for the state. This is in accordance with our modeling choice of having the classification matrix be independent of rewards and goals. Finally, their IL method led to better policy summaries, and using state similarities helped. This mirrors our work in that state-similarity –which in our case can lead to state aliasing– ought to be considered for generating better policies. We also confirm the fact that humans are more effective when following a simpler policy as we show in our human studies.

Recall that one of the effects of state aliasing on policy execution is delayed execution which can bring in time into the problem. Adding time to MDPs is considered in Semi-Markov Decision Processes(SMDP). SMDP considers the case when time between one decision and the next is a random variable. This has some similarities to how we model delay time since the number of delay steps in a state becomes a random variable. However, there are two critical differences. First, the cause of the delay is not due to an explicit action in the policy nor dependent on the current state alone. Rather, the delay is due to state aliasing and subsequent policy confusion. Second, SMDPs are Markovian, whereas this paper addresses a problem in which the actions of states are coupled and so becomes Non-Markovian.

If we ignore the policy execution delay due to confusion and just think about the state-aliasing, then this problem can be seen as a Partially-Observable Markov Decision Process

(POMDP) problem. If we were to tackle that limited version of the problem with POMDP solvers, there is still the issue that we cannot give a POMDP policy to the human; a policy that is conditioned on belief state (likelihood of possible states). We cannot expect the human to track their posterior state likelihoods accurately. POMDP policies can also be defined by histories of observations. If we think of the state that the human inferred as an observation emitted (the human only knows this "observation"), then the policy we return from solving SAMDP is akin to a POMDP policy conditioned on a history of 1 observation.

With respect to the state-aliasing phenomenon in MDP literature, this has been studied in the lens of POMDPs for agents with active-perception capabilities [Whitehead and Ballard 1991],[Tan 1991], [Whitehead and Lin 1995]. In [Whitehead and Lin 1995] the authors call the problem as perceptual aliasing when an internal state maps to more than one external state due to limitations of the sensing process. Their approach of "Consistent Representation"(CR) pulls together the prior work on the perceptual aliasing problem. The common assumption across those works is that a consistent representation of the state that is Markovian can be built from the immediate environment with sensing actions. [Whitehead and Lin 1995] also considered "stored-state" architectures where history was incorporated for state inference. In all of these works, there are additional computation steps required in the policy to improve state detection. These approaches assume the agent has sufficient computational capacity, and consistently infers the correct subsequent state representation; this is necessary since the policy is conditioned on this consistent representation. We do not think one can expect this from humans. So we instead rely on reducing the execution errors by allowing for incorrect sensing (incorrect state classification) by humans.

## Conclusion and Future Work

In this paper, we describe the problems that can arise from state aliasing when humans execute a policy; these are execution errors and delays in policy execution. We formally define the problem of computing policies in SAMDPs and define how delays and errors can be computed for a given policy in the SAMDP using the state classification likelihood. We discuss how the state classification likelihood can be empirically evaluated for human agents and also how domain rewards can be appropriately discounted to account for state detection time. Given the description of SAMDP, we present a modified policy iteration algorithm (GVPI) which searches for policies that account for delays and errors, and optimize the expected value. GVPI also allows searching for simpler policies by increasing a hyperparameter that penalizes policy confusion score. Lastly, we conducted human studies to show how our assumptions translate to real-world behavior.

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
