# OpenReview forum: "Synthesizing Policies That Account For Human Execution Errors Caused By State Aliasing In Markov Decision Processes"
_icaps-conference.org/ICAPS/2021/Workshop/XAIP — XAIP 2021_

### Official Review · Program_Chairs · 2021-07-06
**what's this title business in openreview?**

**Rating:** 9
**Confidence:** 4

**Review:**

sorry but I guess I'm old and conservative.

Anyway, this is a great paper and a perfect match for XAIP.

The authors propose a formal model of an MDP in which states may be confused by a human execution agent. They define optimization objectives trying to avoid such confusion. They conduct both, benchmark studies and a user study, and the results are quite convincing. Overall, a nice piece of work, soundly formalizing and exploring an important aspect of explainability.

I don't have anything to criticise. Maybe the authors could be a bit more explicit from the start (intro) to what degree similar proposals have been made/noit made before (the relate work section mentions related concepts in POMDP contexts, which imho would be better to discuss up front already).

BTW, I wish google maps was taking this kind of thing into account in route planning already! :-) (easier to follow route, where is the right optimization objective when you need it?)

---

### Official Review · AnonReviewer2 · 2021-07-06
**Paper Review**

**Rating:** 9
**Confidence:** 3

**Review:**

Overview: This paper presents a method to incorporate human execution errors and delays within policies to increase robustness of certain planning scenarios.

Comments:

- Motivation for work is clear, and its potential practical use explained well. Relevance to human-ai systems is obvious.

- Problem well-defined; initially the classification likelihood measure seems arbitrary, however this is explained later as potentially requiring an empirical source. While this is not used in the experiments, it's interesting to think of the possible improvements from identifying common misidentified states given the same circumstances under which the system would be used by a user (i.e. time to make a decision, as mentioned).

- Evaluations were thorough, and while the human study was inconclusive w.r.t. incorrect actions, it is still valuable to see the improvement in throughput.

- An interesting use for this approach could also be diagnosing the chokepoints of a human-ai system for feedback to some other explanation system, in the hopes of reducing the classification likelihood factor until the problem nears an MDP.

Overall nice ideas, well-communicated.

Minor Typos:

- In general, I'm not sure if square bracket in-text citations are correct AAAI format (could be mistaken, but I believe they're meant to be of the form: (<Author Last Name(s)>, <Year>))

---

### Official Review · AnonReviewer1 · 2021-07-08
**The use of special case POMDP to account for state-aliasing**

**Rating:** 4
**Confidence:** 4

**Review:**

The paper presents the use of a State-Aliased MDP (SAMDP) model to generate a policy that takes into account the problem of agent’s state-aliasing via the form of input classification likelihood matrix. The notion of execution delay is modeled by including a null action with domain-specific observation dynamic and decaying reward process. A modified policy iteration search is used to compute a “good” policy (sampling-based; so no guarantee on optimality) with dependency on a hyperparameter that controls the weighting between policy simplicity and value. A user study is performed with results supporting the notion that simpler policies lead to increased number of executed actions, which correlates to increased number of correctly executed actions.

Although I recognize the relevancy of state-aliasing towards explainability in planning, I thought the paper lacked in depth of explaining the functional difference from solving a POMDP and that its evaluation methods were weak.

With the dynamics of the null action defined by the domain, state-aliasing or the input classification likelihood matrix is the key factor in this paper that can be modeled by the observation probabilities in POMDP. I thought the statement “we cannot expect the human to track their posterior state likelihoods accurately" was a very strong assumption about the belief updating capabilities of humans which lacked support. With SAMDP being a special case of POMDP with history of 1 observation, the paper should clearly discuss the novelty and the need for SAMDP formulation, and the reason for the GVPI algorithm over various existing POMDP solver methods.

Thus, the evaluation should also compare against policies derived from POMDP solution algorithms, and for fair comparison, the overall task performance (not just the value and confusion score separately) can be investigated as the number of sampling iterations vary. The benefit of policy largely depends on the reward distribution of the domain, so evaluating the efficacy across various problems with different reward structures (some “flattened”, some even but with a number of distributed peaks, and some with very few concentrated peaks like the examples included in the paper) would also help motivate the impact of the framework.

The paper should also discuss how the hyperparameter (w) balancing between the value and confusion should be specified. This acts as a strong bias towards the resulting policy where I think it’s largely dictated by the designer’s intelligent understanding of the input classification likelihood matrix coupled with the reward distribution.

I thought the user study findings (simpler policies lead to increased number of executed actions, which correlates to increased number of correctly executed actions) were uninteresting and did not precisely showcase the efficacy of balancing value and state-aliasing confusion. Thus, evaluating based on the overall task performance and obtaining users’ feedback about their perceived difficulty of policy interpretation coupled with objective task performance would have been more valuable. I also advise against using phrases like “we also confirm that fact that humans are more effective when following a simpler policy..” with statistical tests performed on a small specific domain.

---

### Meta-Review · Area_Chairs · 2021-07-07

**Recommendation:** Accept
**Confidence:** 4

**Metareview:**

Thanks very much for submitting your paper!

Summary: The paper proposes a formal model of an MDP that account for humans' erroneous executions and delays.

Strengths:
- Clear motivation and relevant research topic
- In-depth evaluation with both benchmark and user study

Limitation:
- Little information about similar studies in the introduction

We hope that you find the reviewers' comments to be informative, please take them into account when revising your papers. We look forward to your presentation!

---

### Decision · Program_Chairs · 2021-07-08

Accept